# Plant Leaf Functional Adaptions along Urban–Rural Gradients of Jinhua City

**DOI:** 10.3390/plants13121586

**Published:** 2024-06-07

**Authors:** Chenchen Cao, Shufen Cui, Xinyu Guan, Yuanjian Chen, Yongqi Zhang, Xingwen Lin, Chaofan Wu, Zhaoyang Zhang, Fei Zhang, Yuling Xu, Zhenzhen Zhang

**Affiliations:** 1College of Geography and Environmental Sciences, Zhejiang Normal University, Jinhua 321004, China; caochenchen@zjnu.edu.cn (C.C.); 202120200822@zjnu.edu.cn (X.G.); crossborder@zjnu.edu.cn (Y.C.); zhang-yongqi123@zjnu.edu.cn (Y.Z.); linxw@zjnu.edu.cn (X.L.); cfwdh@zjnu.edu.cn (C.W.); zhzhyang@outlook.com (Z.Z.); zhangfei3s@163.com (F.Z.); 2College of Business, Lishui University, Lishui 323200, China; 471101875@lsu.edu.cn; 3Zhejiang Jinhua Ecological and Environmental Monitoring Center, Jinhua 321000, China; xuyul0906@gmail.com

**Keywords:** summer heat, urbanization, plant survival strategies

## Abstract

Environmental changes induced by urbanization may significantly alter plant survival strategies, thereby introducing uncertainties in their ability to withstand extreme heat. This study, centered on Jinhua City, distinguished urban, suburban, and rural areas to represent the various intensities of urbanization. It examined the leaf function properties of evergreen and deciduous trees common in these regions, focusing on leaf and branch characteristics. Employing an analysis of variance (ANOVA), principal component analysis (PCA), and path analysis (PA) of the plant functional traits and the climatic factors of each region, this study assessed the impact of urbanization on plant survival strategies. By tracking changes in plant functional traits from June to August, it explored the capacity of plants to acclimate to urban-warming-related heat stress across different urbanization gradients. The findings revealed that leaf thickness (LT) and stomatal size (SS) initially decreased and then increased, whereas specific leaf area (SLA) and leaf tissue density (LTD) first rose and then declined, from rural to urban regions. From June to August, branch wood density (WD), chlorophyll (Chl) content, LTD, and leaf dry matter content (LDMC) increased, whereas SLA and leaf water content (LWC) diminished, in all regions. PCA suggested that there was no significant change in the resource allocation strategy of plants (*p* > 0.05), with drought tolerance significantly reduced in the suburbs on the gradient of urbanization (*p* < 0.05). During the summer, with high temperature, plants were predominantly biased towards slow-return, conservative strategies, particularly among evergreen species. Compared to precipitation, PA revealed a significant urban warming effect. During summer, temperature was the main factor influencing resource investment strategy and drought resistance, with a notably stronger impact on the former. The high temperature in summer promoted a conservative survival strategy in plants, and the urbanization effect increased their tolerance to high temperatures.

## 1. Introduction

In recent years, the impact of the urban heat island effect on vegetation has received much attention due to China’s rapid urbanization [1]. Temperature plays a key role in plant growth. Urbanization-induced warming, on the one hand, probably benefits plant survival by contributing to net productivity accumulation and thus promoting growth and development [2]. Evidence can be found of significantly increased plant productivity with decreasing elevation in previous studies [3,4,5]. On the other hand, combined with summer heatwaves, extreme high temperatures in urban areas expedite plant water loss and individual growth [6]. In some cases, temporary or permanent leaf wilting will occur [7]. At the same time, the expansion of impervious surfaces and the development of sophisticated drainage systems in urban areas have substantially reduced the soil water retention capacity, exacerbated soil drought conditions, and posed a significant threat to the survival of urban vegetation [8]. Therefore, finding a balance between maintaining growth and ensuring survival becomes a paramount challenge for plants to survive in urban environments.

The adaptation of plant survival trajectories, particularly through modifications in leaf functional traits, is quite critical to their acclimation to urban environments [9]. Given that leaves are the primary basis of plant growth, and are highly sensitive to environmental changes [10], plants usually exhibit a variety of response mechanisms under different conditions. In favorable environments, plants tend to have a higher specific leaf area (SLA), nitrogen content, and photosynthesis rate, but shorter lifespans, which is characterized as a “fast investment-return” or extensive investment strategy [2,11]. In contrast, under more challenging conditions, plants tend to show a marked reduction in SLA and photosynthetic rate, coupled with an increase in leaf lifespan. This approach is known as a “slow investment-return” or conservative investment strategy [2,11]. The former facilitate faster growth but offer less adaptability and tolerance to extreme conditions, whereas the latter favor plant survival in adverse environments at the cost of reduced growth rates [11,12]. Therefore, it is hypothesized that urban warming might prompt plants to adopt a “fast investment-return” survival strategy. However, the long-term sustainability of this strategy remains uncertain in the midst of the extreme heat stress frequently experienced during summer, especially given the imperative need for the plant to maintain hydraulic safety.

In addition, plant responses to urbanization may be inconsistent across different plant life forms. Evergreen and deciduous tree species have leaves with distinct life forms and resource allocation strategies. Generally, evergreen species typically follow a ‘slow investment-return’ strategy, while deciduous trees follow a ‘fast investment-return’ approach [13]. It has been thought that deciduous species would be more sensitive to rising temperatures considering the changes in the SLA, net photosynthetic rate (A), leaf nitrogen content, chlorophyll (Chl), and stomatal conductance [14]. In turn, increased precipitation leads to shorter leaf lifespans and thinner leaves in deciduous species, combined with a significant increase in photosynthetic capacity, a pattern not observed in evergreens. Consequently, deciduous plants may be more responsive in urban heat island conditions [15]. However, this hypothesis has not been well tested.

In previous studies, spatial comparison approaches (urban–rural gradients) are mostly used to study the temporal trends of urbanization’s effects on plant response [16]. This study focused on Jinhua City and employed gradients from urban to suburban and rural areas to represent varying levels of urbanization. Evergreen and deciduous tree species common in these regions were selected to examine their responses to leaf functional traits. Thereby, the impact of urbanization on plant survival trajectories was systematically assessed, particularly during summer heat and drought periods. This study primarily (1) addresses the variation in plant resource allocation strategies along the urbanization gradient, and (2) presents a comparative analysis of resource allocation strategies among different plant life forms in response to urbanization.

## 2. Results and Analysis

### 2.1. Response of Plant Functional Traits to Urbanization Effects

The differences in plant functional traits under the urbanization gradient in June were analyzed by three-way ANOVA (Appendix A). Decreased LT was observed from 0.24 mm in rural regions to 0.22 mm in suburban regions (*p* < 0.05), but no significant difference was found between suburban and urban areas (Figure 1h, *p* > 0.05). SS demonstrated a reduction from 4.78 μm^2^ in rural locales to 3.9 μm^2^ in suburban settings, followed by an increase to 4.7 μm^2^ in urban environments (Figure 1g). Conversely, an initial increase and a subsequent decrease in SLA and LTD from rural to urban regions were noted (Figure 1b,i). The same patterns were only observed for deciduous species, while evergreen species exhibited a decreased SS from 5.7 μm^2^ in rural areas to 4.5 μm^2^ in suburban regions, before rebounding to 5.0 μm^2^ in urban settings (Figure 1g). The other traits displayed no significant regional disparities (Appendix A and Figure 1). Additionally, we also performed an ANOVA on these traits of six key species along the urban–rural gradient. When a specific species was considered, most trait variations along the urban–rural gradient showed the same pattern as when all the species were considered, even though some traits had different responses (Appendix A).

### 2.2. The Impact of Summer Heat on Plant Functional Traits

Compared to June, WD, Chl, and LDMC exhibited a significant increase in August. However, this trend gradually decreased from rural to urban areas, indicating reduced sensitivity of urban plants to high temperatures for these functional traits (Figure 2). Similarly, LTD showed a noteworthy increase, particularly in suburban regions (Figure 2b). On the contrary, LWC and SLA significantly declined in August, with the degree of decrease diminishing from rural to urban areas, suggesting lower heat sensitivity in urban plants. SD generally increased in August, with more pronounced changes seen from rural to urban areas (Figure 2d). LT only experienced a significant decrease in suburban locations (Figure 2h). No significant changes were observed in other plant functional traits (Figure 2e,g).

Trait variations from June to August in evergreen species showed no significant differences across rural and urban areas for WD, Hv, SD, and SS (Figure 2a,e,d). However, the differences for SLA and LWC significantly increased (Figure 2i,j), while those for LDMC and Chl decreased noticeably, from rural to urban regions (Figure 2c,f). Notably, there were no distinct trait differences between suburban and urban areas. Additionally, LTD and LT exhibited an initial increase followed by a decrease from rural to urban areas during this period (Figure 2b,h).

For LT and SLA, the June–August variability in deciduous species did not show significant differences across the three regions (Figure 2h,i). WD, LTD, LDMC, SD, Hv, and Chl exhibited significant decreases from rural to urban areas (Figure 2a–f). Specifically, WD, LDMC, SD, and Chl did not display significant differences between suburban and urban areas (Figure 2a,c,d,f). This indicated that the plant functional traits of deciduous tree species in urban areas are not sensitive to the impact of high temperatures. There were no significant differences in WD, LTD, Hv, and Chl from rural to suburban areas (Figure 2a,b,e,f).

Additionally, we conducted a one-way ANOVA on the changes in plant functional traits for two deciduous and four evergreen tree species that co-exist along the urban–rural gradient from June to August. The results indicated that the trends in plant functional trait changes for the six individual tree species along the urban–rural gradient were relatively consistent with the overall trend (Appendix A). This once again verifies the accuracy of the above conclusions.

### 2.3. Urbanization-Driven Changes in Plant Resource Allocation Strategies

The PCA revealed that the first two principal components account for 39.3% (PC1) and 22.6% (PC2) of the variance (Figure 3a). PC1, characterized by high LDMC and LTD on the positive axis along with that of LWC, SLA, and Hv on the negative axis, closely aligns with the “Leaf Economic Spectrum (LES)” and its “investment-return” strategy axis. Individuals that load on the positive and negative sides of PC1 could be interpreted as embodying a “slow investment-return” (conservative) and a “fast investment-return” (expansive) strategy. PC2 was oriented with the highest LT and Chl loads on the positive side and the lowest SLA and SD loads on the negative side, revealing a trade-off between strong and weak drought tolerance.

There was no significant difference in plant loading fraction on PC1 from rural to urban areas, while it decreased initially and then increased on PC2 (Figure 3b). For the evergreen species, the load on PC1 increased steadily from rural to urban areas, while the load on PC2 decreased progressively (Figure 4a). In contrast, deciduous species showed no significant change in their PC1 load from rural to urban areas, while their load on PC2 initially decreased and then increased (Figure 4b).

From June to August, there was a notable shift in plant loading on PC1 towards the positive side, and this shift exhibited a decreasing trend from rural to urban areas (Figure 3b). The difference between plant loads on PC2 in August and June was not significant for most regions, except for suburban areas, where PC2 loads were significantly lower in August than in June (Figure 3b).

When considering the living forms, there was a progressive increase in loading from rural to urban areas on PC1 in June for evergreen species, while this trend was reversed in August, indicating a reduction in variation from rural to urban areas on PC1 (Figure 4a). In contrast, deciduous plants showed no significant difference in PC1 loading across regions in June, but decreased from rural to urban areas in August, presenting a reduced PC1 difference between June and August towards urban areas (Figure 4b). PC2 loading for evergreen species did not show significant variation across regions for both June and August (Figure 4a). For deciduous species, the loading on PC2 decreased and then increased along the urban–rural gradient in June and August (Figure 4b). When comparing June and August, there was no significant difference across the city for evergreen species, but for suburban deciduous species, the loading in August was significantly lower than in June (Figure 4).

### 2.4. Driving Factors of Plant Resource Allocation Strategies

The path analysis revealed that urbanization exerted a positive impact on temperature and a negative impact on precipitation, particularly during the periods of March to June and July to August (with particular emphasis on the latter period, Figure 5). In the months of March to June, neither temperature nor precipitation demonstrated any influence on PC1 and PC2 (Figure 5a). However, during the months of July to August, a contrasting relationship was observed: temperature exhibited a negative correlation with PC1 and a positive correlation with PC2 compared to the previous period (March to June) (Figure 5b). Furthermore, precipitation displayed positive associations with both PC1 and PC2.

## 3. Discussion

### 3.1. The Impact of Urbanization on Plant Functional Traits

In our study, the benefits of warming on plant growth capacity proposed in previous studies were indeed proven to some extent with regard to urban warming [17]. SLA and LTD increased to enable enhanced light capture and growth rate capacity from rural to suburban regions (Figure 1b,i), whereas LT and SS decreased to facilitate stomatal regulation more efficiently, which maximized the gas exchange rates [14] (Figure 1g,h). These changes proved that plants have weaker drought resistance in suburban regions then other regions.

However, these benefits tend to be suppressed in the urban center, considering the decreased SLA and LTD, as well as increased LT and SS (Figure 1b,g–i). It was revealed that there was a reduction in pants’ growth and photosynthetic rates and an increase in their resistance to drought stress. Moreover, the overarching resource investment strategies of plants also remained relatively consistent across the urban–rural gradient (Figure 3b), and temperature had no relation to the PC1 changes along the urban–rural gradients (Figure 5). This observation stands in contrast to previous research, which argued that urban environments enhance the growth of vegetation [18].

This discrepancy could largely be attributed to the unique environmental stresses associated with urbanization [9]. Urban environments are characterized not only by higher temperatures, but also by a myriad of other stress factors, including air pollution, soil compaction, light pollution, etc. These conditions might compel plants to adopt more conservative resource allocation strategies. A previous study found that *C. camphora* displayed conservative hydraulic and fast-return economic strategies to acclimate to urban hot and dry environments [19], as a means to cope with challenging urban environments. For instance, exposure to urban pollution often leads to leaf thickening and a decrease in photosynthetic capacity, which, in turn, slows down plant growth [20].

### 3.2. Changes in Plant Regulatory Capabilities under High Summer Temperatures

Even though urbanization has a weak influence on plant resource allocation strategies, in the face of summer heat waves, plants tended to be more adapted to the urban environment. Prior research argued that stable plant leaf functional traits under extremely high temperatures serve as an indicator of their environmental adaptability [21,22]. Our study found that under the influence of high temperatures, trait variations between June and August for WD, Chl, LDMC, LWC, and SLA gradually decreased from rural to urban areas. Meanwhile, urban regions exhibited the lowest PC1 variability (Figure 3b). These findings suggested that urban plants possess higher adaptability to sustain a more stable growth state in response to high temperatures [23]. This indicates plant plasticity, whereby plants adapt to changes in the living environment.

Some studies have suggested that this could be associated with increased plant resistance to drought due to the hot and arid urban climate [24]. However, our results did not support this possibility, since the loading of PC2 between rural and urban areas (Figure 3b) indicated remaining plant drought tolerance. This stability may be attributed to urban tree growth trajectories being better suited to high-temperature conditions, coupled with the complexity of growth-driving factors for urban plants. A study found that urban environments promote plant resistance to high temperatures [13], which might be associated to diverse planting methods, soil types, and fertilization practices, each significantly influencing plant growth [25].

### 3.3. Variations in Urban Responses among Different Life Forms of Plants

Our study indicated that plant functional traits, reflecting warming benefits in suburban regions, were mostly contributed by deciduous species. These species exhibited the same pattern of LT, SS, SLA, and LTD variation as when all species were considered along the urbanization gradients (Figure 1b,h–j). In contrast, evergreen species only showed varied SS [23] (Figure 1g). Accordingly, PC2 related to drought tolerance for evergreen species was also maintained from rural to urban areas, in contrast to the decreased PC2, particularly in suburban areas for deciduous species (Figure 4a). These results demonstrated that deciduous species were more sensitive to urban warming, as we proposed above [26].

These differences could be attributed to their distinct resource allocation strategies [13]. Our research demonstrated that evergreen tree species adopted a more conservative resource allocation approach (Figure 4a), progressively favoring structural components as they transitioned from rural to urban areas. This adaptation allowed them to better withstand urban challenges [13,27]. In contrast, deciduous species, located on the resource acquisition end of the spectrum, demonstrated increasing investment in resource acquisition as they moved from rural to urban environments (Figure 4b). This suggested that in urban climates characterized by hot and dry conditions, deciduous species may adopt a “riskier” strategy focused on maintaining high growth efficiency, albeit potentially at the expense of reduced resistance to environmental stressors [28,29].

It is worth noting that despite their different survival trajectories, both evergreen and deciduous plants tended to adopt more conservative resource allocation strategies in high-temperature environments. This shift toward conservatism was most prominent in rural areas and gradually decreased toward urban areas (Figure 4). This trend suggested that increased structural investment becomes crucial for both types of plants to cope with higher summer temperatures. A higher LDMC played a significant role in this adaptation by reducing water loss from the leaf surface [30]. As a response to water deficits caused by high temperatures, plants tended to develop thicker leaves and lower SLA [24]. These adaptations improved water use efficiency and enhanced the ability of plants to survive in challenging conditions [31].

### 3.4. Limitations of the Study

Our study offers valuable insights into how plants adapt to urbanization, but it is important to acknowledge some limitations that could introduce uncertainties to our findings. Firstly, we did not account for the influence of plant size. Research suggests that larger plants are less sensitive to increasing temperatures, making them more resilient in extreme conditions [32]. Secondly, this study only focused on the differences in the functional characteristics of evergreen and deciduous plants along the urban–rural gradient, which ignored some more vulnerable functional taxa, such as shrubs and grasses, which are less resilient in urban environments. They often struggle to recover from high temperatures and drought [13,33]. Furthermore, we assumed that natural environmental factors were the main influence on plants. However, in urban areas, city irrigation can significantly reduce plant stress induced by heat and dry spells. Suburban areas with less frequent irrigation may show different plant responses to drought stress due to varying watering patterns [13]. In conclusion, future research should consider diverse factors to better grasp vegetation dynamics amidst urbanization.

## 4. Conclusions

In our study assessing the effects of urbanization on plant growth, we analyzed the functional traits of various plant life forms across an urbanization gradient. Our findings revealed the following:

(1) Benefits of urban warming on plant growth capacity were only observed in suburban regions, with a price of decreased drought tolerance. Meanwhile, the plant resource investment-return strategy was not markedly altered along the urban–rural gradient.

(2) After a summer heatwave, the plant resource investment-return strategy shifted towards more conservative approaches. Urban plants experienced less of a shift during this period, which revealed their effective adaptation to heat stress.

(3) The plant functional traits of deciduous species demonstrated greater sensitivity to urbanization compared to the weak traits and PC1 variations along the urban–rural gradient for evergreen species. However, under summer heatwaves, both evergreen and deciduous plants tended to adjust their resource investment-return strategies towards more conservative methods.

## 5. Materials and Methods

### 5.1. Study Area

This study was conducted in Jinhua, a city in the heart of Zhejiang province that occupies the eastern part of the Jinqu Basin (Figure 6a). During the past few decades since 1995, Jinhua City has witnessed rapid urban expansion, with an net increase of 440 km^2^ in the build-up area [34]. The region is characterized by a subtropical monsoon climate marked by distinct seasonal variability. It receives annual average precipitation of 1593.8 mm and maintains an average annual temperature of 19.1 °C. The local vegetation straddles the ecotone between the subtropical evergreen broad-leaved forests and the temperate deciduous broad-leaved forests, resulting in a rich diversity of plant species. It is worth noting that according to the criteria of the China Meteorological Administration (daily maximum temperature ≥35 °C, https://www.cma.gov.cn (accessed on 21 March 2022)), a total of 59 days were classified as extremely hot days (EDHs) during the study period. The maximum daily temperature in Jinhua City in summer reached 40 °C (usually during 11:00–13:00), indicating that plants in Jinhua City experienced extremely high temperatures in summer, with the average monthly temperature in urban areas reaching 37.4 °C.

### 5.2. Assessment of the Urbanization Effect on Climate

To assess the impact of the urban heat island effect on the climate and vegetation of Jinhua City, MODIS MOD11A2 datasets were collected for the period spanning 2016 to 2020. These datasets provided land surface temperature products with an 8-day temporal resolution and a 1 km spatial resolution. Studies have indicated that land surface temperature (LST) is highly correlated with air temperature [35]. Thus, in this study, LST was used to quantify the UHI effects with a high spatial resolution. These data facilitated the generation of detailed spatial distribution maps of UHI effects in Jinhua City (Figure 6a). Meanwhile, Chinese Monthly Precipitation Datasets (1901–2020) with a spatial resolution of 1 km for the period spanning from 2016 to 2020 were collected over Jinhua City [36]. These datasets enabled the creation of comprehensive maps illustrating spatial variability in precipitation across the city (Figure 6b,c).

To quantify the urban heat island effect, a total of 10 buffer zones were set up every 3 km from the central urban area (Figure 6a). We calculated the annual mean temperature and precipitation across each of these 10 zones (Figure 6d). A significant urban heat island effect within a 12 km radius of the city center was obtained (Figure 6e,f), and the LST differences between urban and rural areas reached 5.7 °C in summer and 3.8 °C in spring, which were similar to those reported in previous warming experiments (usually 2–6 °C) [37,38]. Considering that LST is generally higher than air temperature (which is directly related to plant functions), we extracted the mean air temperature (T_a_) of the state control station in Jinhua City in 2022 (29°11′ N, 119°66′ E) obtained from the National Meteorological Science Data Center (http://www.nmic.cn/ (accessed on 2 March 2022)). These data were fitted to the monthly mean LST (Appendix A). According to this relationship, the urban–rural difference in T_a_ was predicted to be 5.6 °C in 2022. Thus, the temperature difference in this study was sufficient to cause the adapted regulation of plant functional traits. Interestingly, the urban rain island effect was observed only within a 5 km radius outside the urban core (Figure 6e,f). On a seasonal basis, temperature peaks were consistently recorded in July and August, and precipitation peaks were recorded in June, across all buffer zones (Figure 6d). To differentiate the varied impacts of urbanization on climate across seasons, we computed changes in precipitation and temperature along the urbanization gradient from March to June and from June to August (Figure 6g,h).

### 5.3. Selection of Sampling Sites and Tree Species

Ten sampling sites across the temperature gradients were identified and selected in the north of Jinhua City (Figure 6a). An increment of 2 °C was employed as the key criterion to divide the city into rural, suburban, and urban areas. Consequently, sites such as Wu Zhou Park, Zhejiang Normal University, Huan Dong Park, and Wu Xing Park were classified as urban areas (1–4). Jian Feng Mountain and Zhi Zhe Temple were identified as suburban areas (2–6). Shuang Long Power Station, Shuang Long Cave, Lu Nu Lake, and Huang Da Xian Ancestral Temple were designated as rural locations (7–10) (Figure 6b,c).

In order to minimize the disturbance to the environment, the selected tree species were mostly located at the core of each park, with less interference from human activities. The roots of the plants were relatively well developed, growing in the uniform local yellow and red soil. The water for these trees was mainly provided by precipitation, and watering was applied only when the trees withered. We finally collected a diverse selection of mature woody plants (nine evergreen plant species and sixteen deciduous plant species) that are commonly found in urban areas of subtropical regions (Table 1). All these trees were mature with good growth conditions.

It should be emphasized that we did not use exactly the same tree species at each sample site. On the one hand, it is hard to guarantee that all the sites have the same species. On the other hand, when the same species were selected, the response of one of few species will probably lead us to draw a biased conclusion. Therefore, in order to more truly reflect the response of vegetation in nature, the dominant species in each region were selected. This method has been widely adopted in previous studies to investigate vegetation’s responses under various environmental gradients, such as drought, elevation, precipitation, etc. [3,13].

### 5.4. Sample Collection and Metric Determination

The SLA, leaf thickness (LT), leaf dry matter content (LDMC), leaf water content (LWC), Huber value (Hv), branch wood density (WD), leaf tissue density (LTD), Chl, stomatal size (SS), and stomatal density (SD) of plants were selected in our study according to a previous study [10].

The sample collection and assessment of plant functional traits were carried out during June and August 2022. In June, fully mature leaves facilitate the quantitative assessment of the impact of urbanization on plant resource allocation strategies [19]. In August, leaves exposed to the summer heat wave (32–38 °C in 2022, Figure 6h) allow for the evaluation of the adaptability of plants in different regions to high-temperature conditions [39]. Four to five evergreen and deciduous tree species were selected at each sampling site. Sunlit branches were harvested using a 10 m pole pruner from three disease-free, unshorn, well-grown individuals from each species. One branch was cut from each individual. The branches were immediately enclosed in black plastic bags to minimize moisture loss and brought to the laboratory in a heat box for trait measurements.

All collected branches (*n* = 450), and fresh leaves were immediately weighed to determine their fresh weight (W) using an electronic balance with a precision of 0.0001 g. The leaves were immersed in water to minimize the disturbance of the effect of leaf evaporation on the leaf thickness. LT was measured at the upper, middle, and lower sections using a vernier caliper with a precision of 0.01 mm (DL92150, Deli group Ltd., Ningbo, China), avoiding major veins, with the average of these measurements representing the leaf’s actual thickness (*n* = 450). The leaves were digitally scanned using an Epson V19 scanner (Epson (China) Co., Ltd., Beijing, China) and analyzed with a Win-RHIZO (Pro 2009) system (Regent Instruments, Québec, Canada) to determine the leaf area (LA) (*n* = 450). Leaf volume (V) was then calculated by multiplying LT by LA. Subsequently, the leaves were oven-dried at 65°C until a constant weight was achieved (typically within 72 h), and their dry weight (*W*_l_) was measured using the same precision balance (*n* = 450). Key leaf traits, including SLA, LTD, LDMC, and LWC, were computed using Formulas (1)–(4), respectively.
(1)$$SLA=\frac{LA}{W_{l}}$$
(2)$$LTD=\frac{W_{l}}{V}$$
(3)$$LDMC=\frac{W_{l}}{W}$$
(4)$$LWC=\frac{W-W_{l}}{W_{l}}\times 100\%$$

Chl in the leaves was quantified using a handheld chlorophyll meter (Yaxin-1260, Beijing Yaxin Liyi Technology Ltd., Beijing, China). Three to five leaves were selected for each branch (*n* = 450). For each leaf, ten distinct measurements were taken across various locations, carefully avoiding the primary veins of the leaves, and the mean of these readings was calculated to represent the Chl value for that specific leaf.

SD and SS were determined using the nail polish imprinting method. SD was measured under 40× magnification, while guard cell width (w) and length (L) were assessed under 100× magnification. SS was then calculated as the product of guard cell length and the cumulative width of two guard cells (2w).

A 10 cm segment from the base of each branch (*n* = 450) was peeled to determine their volume (V_t_) with the water displacement method. The branches were then oven-dried at 65 °C until a constant weight was reached, typically within 72 h, to measure their dry weight (W_b_). Vt and W_b_ were used to calculate the WD (Formula (5)). The basal diameter of the peeled stem segment was measured using a vernier caliper, allowing for the calculation of the basal stem’s cross-sectional area (S_t_). The Hv was then derived using Formula (6).
(5)$$WD=\frac{W_{b}}{V_{t}}$$
(6)$$Hv=\frac{S_{t}}{W_{l}}$$

### 5.5. Data Processing and Analysis

To investigate the diverse responses of urban plants to urbanization, first of all, we tested the normality of the variable data, and all of them conformed to a normal distribution. Then, a three-way ANOVA was used to examine the significant differences in plants’ functional traits on urban–rural gradients using SPSS 26.0 (IBM SPSS Inc., Chicago, IL, USA), to assess the effect of urbanization and its interaction with life forms and sampling period on leaf functional traits.

Principal component analysis (PCA) was used to explore the differences in resource allocation strategies under the urbanization gradient. PCA could standardize the data with different orders of magnitude, thus enabling the comparison of plant functional traits among different regions with different species compositions. Each individual’s load score was subtracted from each principal component to assess the strategy they adopted.

Furthermore, to identify the urban-warming-related plant functional adaptations and interruptions in precipitation, path analysis was carried out on the LST and precipitation, as well as the mean loading score for each sample site, using Amos software (Amos 27.0, IBM SPSS Inc., Chicago, IL, USA). Model parameters were estimated using the maximum likelihood estimation method, resulting in a path model with corresponding path coefficients.

Graphical representation of all data analysis results was accomplished using Origin Lab (Origin 2021, Northampton, MA, USA).

## Figures and Tables

**Figure 1 plants-13-01586-f001:**
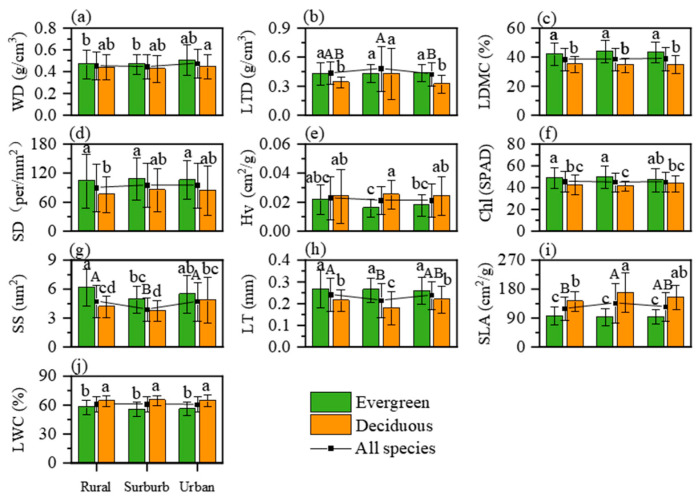
Comparison of functional traits of evergreen and deciduous tree species in different regions in June. Different capital letters indicate significant differences in functional traits of all living plants (*p* < 0.05), and different lowercase letters indicate differences in functional traits of evergreen and deciduous tree species. Where (**a**–**j**) represent WD, LTD, LDMC, SD, Hv, Chl, SS, LT, SLA and LWC respectively.

**Figure 2 plants-13-01586-f002:**
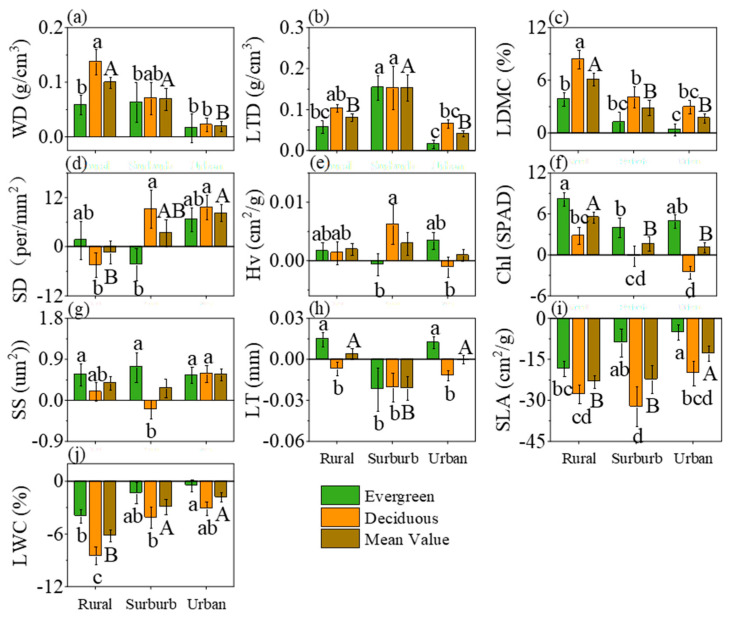
Plant functional trait differences from June to August along the urbanization gradient. Different capital letters indicate significant differences in functional traits of all living plants (*p* < 0.05), and different lowercase letters indicate differences in functional traits between evergreen and deciduous tree species. Significant (*p* < 0.05). Where (**a**–**j**) represent WD, LTD, LDMC, SD, Hv, Chl, SS, LT, SLA and LWC respectively.

**Figure 3 plants-13-01586-f003:**
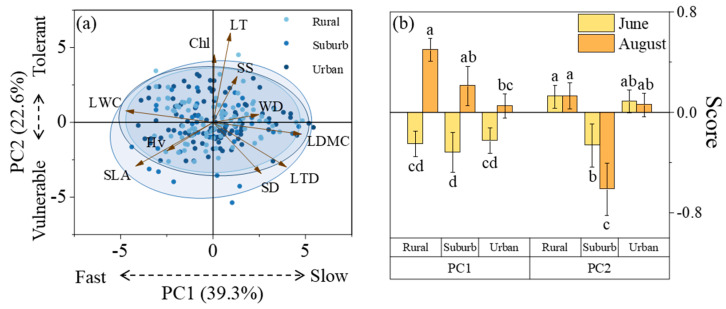
Principal component analysis diagram (**a**) and principal component score diagram of plant functional traits (**b**). Lowercase letters indicate significant differences (*p* < 0.05) in functional traits of evergreen tree species and deciduous tree species.

**Figure 4 plants-13-01586-f004:**
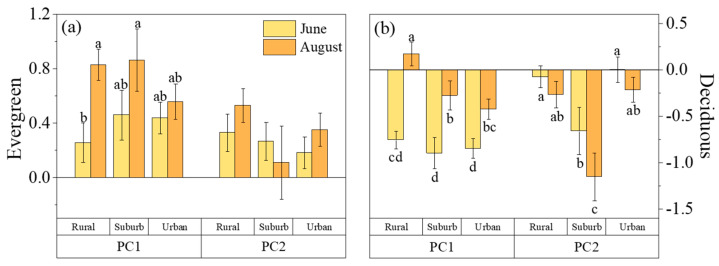
Principal component score plot of evergreen tree species (**a**) and principal component loading plot of deciduous tree species (**b**) in June and August. Lowercase letters indicate significant differences in functional traits between evergreen tree species and deciduous tree species (*p* < 0.05).

**Figure 5 plants-13-01586-f005:**
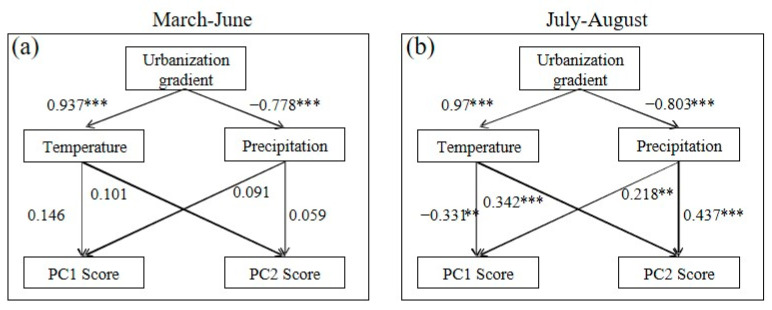
Path analysis for March to June (**a**) and July to August (**b**). *** and ** represent significance levels of 1% and 5%, respectively.

**Figure 6 plants-13-01586-f006:**
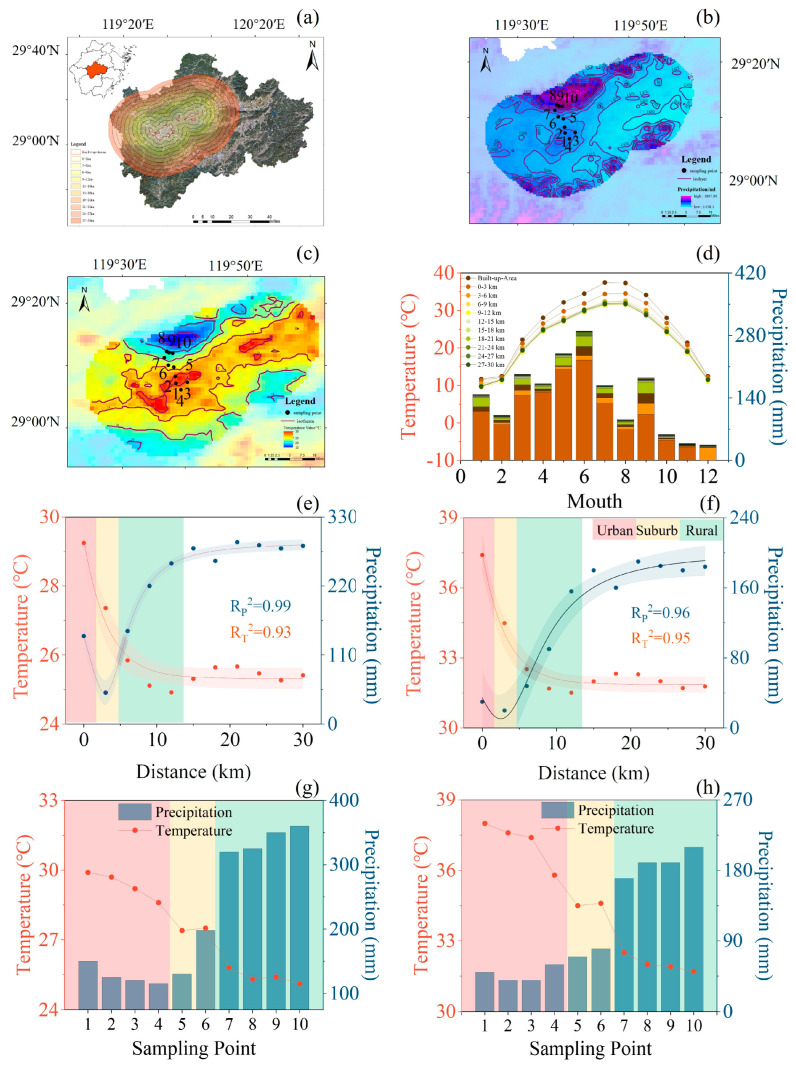
Spatial distribution map of the buffer zone in Jinhua City (**a**), temperature and precipitation contour maps of the buffer zone (**b,c**) and their mean temperature values across each buffer zone from January to December (**d**), monthly mean temperature and precipitation variations across the buffer zone for periods of March to June (**e**) and July to August (**f**), monthly mean temperature and precipitation variations across the 10 sampling sites for periods of March to June (**g**) and July to August (**h**) during 2016–2020.

**Table 1 plants-13-01586-t001:** Sampled tree species within the study area.

Number	Species	Family Name	Life Form
1	*Cinnamomum camphora*	*Lauraceae*	Evergreen
2	*Liquidambar formosana*	*Hamamelidaceae*	Evergreen
3	*Elaeocarpus decipiens* Hemsl	*Elaeocarpaceae*	Evergreen
4	*Osmanthus fragrans* (Thunb.) Loureiro	*Oleaceae*	Evergreen
5	*Ligustrum lucidum* Ait	*Oleaceae*	Evergreen
6	*Ginkgo biloba* Linn	*Ginkgoaceae*	Deciduous
7	*Prunus cerasifera* Ehrhart f. *atropurpurea (Jacq.)* Rehd	*Rosaceae*	Deciduous
8	*Quercus rubra* L.	*Fagaceae*	Deciduous
9	*Populus alba*	*Salicaceae*	Deciduous
10	*Prunus subg. Cerasus* sp.	*Rosaceae*	Deciduous
11	*Liriodendron chinense* (Hemsl.) Sarg	*Magnoliaceae*	Deciduous
12	*Magnolia denudata* Desr	*Magnoliaceae*	Deciduous
13	*Photinia serrulata* Lindl.	*Rosaceae*	Evergreen
14	*Armeniaca mume* Sieb	*Rosaceae*	Deciduous
15	*Cercis chinensis* Bunge	*Leguminosae*	Deciduous
16	*Hovenia acerba* Lindl	*Rhamnaceae*	Deciduous
17	*Sapindus mukorossi* Gaertn	*Sapindaceae*	Deciduous
18	*Fortunella margarita* (Lour.) Swingle	*Rutaceae*	Evergreen
19	*Ilex latifolia* Thunb	*Aquifoliaceae*	Evergreen
21	*Malus spectabilis* (Ait.) Borkh	*Rosaceae*	Deciduous
22	*Celtis sinensis* Pers	*Ulmaceae*	Deciduous
23	*Michelia chapensis* Dandy	*Magnoliaceae*	Evergreen
24	*Lagerstroemia indica* Linn	*Lythraceae*	Deciduous
25	*Diospyros kaki* Thunb	*Ebenaceae*	Deciduous
26	*Sapium sebiferum* (Linn.) Roxb	*Euphorbiaceae*	Deciduous

## Data Availability

The raw data supporting the conclusions of this article will be made available by the authors on request.

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
