# Peer review of "Plant Leaf Functional Adaptions along Urban–Rural Gradients of Jinhua City"

_plants, 2024, doi:10.3390/plants13121586_

Round 1
Reviewer 1 Report
Comments and Suggestions for Authors
This is research that may be important as it deals with the effects that urbanization may have on a species composition (probably urban dominant) including evergreen and deciduous, suggesting a study of plant plasticity. These types of studies have already been carried out previously using altitudinal gradients to determine the effect of climate change on different plant species. However, the study is only aimed at determining effects on deciduous and evergreen trees; it would have been desirable to track one or several species present throughout the urbanization gradient to determine true functional changes of the different species, since it encompasses species in two functional groups may hide specific responses.
It must be taken into account that the urban-rural thermal gradient is very small.
There is also a large amount of literature on altitude changes and the authors did not take them into account, they only dedicated themselves to mentioning literature from this century, as if the 20th century did not exist.
In particular, it is recommended that the authors include key species of the 15 selected species in the study and discuss the differences and similarities together with the functional groups.
Other concerns
Line 25: LTD, what is this?
Lines 34 and 35: How do you know that? Plants are plastic, that is, plants have a thermal tolerance range that can vary and although there are heat waves, the temperature can be in that tolerance range.
Line 53: Plants are not strategist, plants present mechanisms instead of strategies.
Lines 87 and 88: But analyses were just species divided into deciduous and evergreen. Is this analysis sufficient to determine the effect of urbanization on vegetation dynamics?
Line 96: What is a heat day?
Line 103 and 104: Then, this is NOT urban heat island (UHI) but surface urban heat island (SUHI). UHI is by definition air temperature, not surface.
Line 109: buffering with respect to what?
Line 121: 10 must be changed to Ten, please see english spelling rules.
Figure 1:
1d Precipitatión must be represented by bars and and not continuously, since these are accumulated.
Line 134: temperal? Do not abbreviate Jan. Dec. etc.
Line 137: I think it would be better to indicate the sampling sites wit numbers instead of abbreviations, in addition to placing them according to the gradient, from the city center to the rural area. With this, readers would have a better understanding of the figures.
Line 138: why during in bold?
Table 1:
Line 5 of table 1, Ligustrum lucidum is an evergreen species, is missing
Line after 6 on table 1, number is missing?
Line after 7 on table 1: atropurpurea???
Check carefully this table 1
Line 141: Specific leaf area (SLA) is missing
Line 142 chlorophyll (Chl)?
Line 143: stomatal area or stomatal density? The area of ​​a stoma can vary depending on its opening.
Lines 149 and 150: So in the city-rural gradient the same species was not always studied?
Line 156: What accuracy?
Line 157: What kind of vernier? What about evaporation?
Line 160: Leaf area is typically represented as LA (or LA)
Line 172: Make, model and manufacturer is missing.
Equation (6) What is W0? Should it be Wt?
Line 190: Does the data come from a normal population? Did you do any statistical test before applying the ANOVA?
Line 208: write the instead of diverse.
Line 212: (P<0.05) F value? with what test were the differences determined? Include that value along with P.
Line 379: Limitations of the study: Another limitation is not having followed a couple of species or more species present throughout the urbanization gradient, to know the response by species, most likely what is indicated in this research is only a trend between two functional groups.
Comments on the Quality of English Language
Typo errors
Author Response
Many thanks to the reviewer for your questions on our manuscript, which greatly improved the quality of our manuscript. In this revision, we have made detailed revisions to the manuscript in accordance with your comments, and hope that this revision can be approved by you.

Reviewer 2 Report
Comments and Suggestions for Authors
The research described in the manuscript is an attempt to characterize the impact of urban environment on plant surviving strategies expressed as leaf anatomic and functional adjustment. The collected data are abundant and comprehensively analysed. The paper is appropriate for publishing in Plants. However, there are some methodological inaccuracies in description of sampling collection: there is not explained what was the age of the trees, what were the soil conditions (good “garden” soils or urban soils mixed up after building construction) and the growth conditions for roots (lawn or pavement pits), were the sampled trees watered etc. The Authors mention about some issues in the Chapter 4.4. “Limitations of the Study”. Nevertheless, some more information on the sites where tree grew should be given in M&M: on irrigation (if it was performed) and root growth conditions.
These points should be corrected or explained:
1. Chapter 2.3.”Selection of Sampling Sites and Tree Species” should be supplemented with information about approximate tree age (young, mature, old?) and site conditions, as mentioned above.
2. Line 143: stomatal area (SA) was not shown in Figures and described in Results.
3. Line 163: W0, “0” should be as a subscript.
4. Figure 2and 3, text line 230: the parameter SD was not introduced in M&M.
5. Line 221 and 222 in Figure 2 caption: “indicate” instead of “indicated”.
6. Lines 267-271: I can’t see in which Figure or Table this information is provided, please indicate any illustration of these findings.
7. Lines 320-321: for this statement references are necessary.
8. References: items 2, 4, 5, 9, 12, 13, 15, 18, 21, 23, 26 and 30 require some improvement, particularly italics for Latin names and capital letters for names of states, cities, botanists and Latin names of plants.
Author Response

(The authors gave the same response as above.)

Round 2
Reviewer 1 Report
Comments and Suggestions for Authors
After reviewing the new version it is noted that the manuscript has improved significantly, however there are still some concerns as noted below:
Question: However, the study is only aimed at determining effects on deciduous and evergreen trees; it would have been desirable to track one or several species present throughout the urbanization gradient to determine true functional changes of the different species, since it encompasses species in two functional groups may hide specific responses.
Response: We chose evergreen and deciduous tree species because the study area is located in the transitional zone between temperate and subtropical zone. Evergreen and deciduous tree species coexist in this region. We had stressed this opinion in the main text :“We finally collected a diverse selection of mature woody plants (nine evergreen plant species and sixteen deciduous plant species) that are commonly found in urban areas of subtropical regions.”(Lines 148-150). This not an answer to my question, I insist on presenting at least two deciduous and evergreen species to know the effect of urbanization on each.
Question: It must be taken into account that the urban-rural thermal gradient is very small.
Response: Regarding the small temperature gradient between urban and rural areas, we had added this part in the main text: “ and the temperature difference between urban and rural areas reaches 5.7℃ in summer and 3.8℃ in spring, which was similar to those reported in previous warming experiments (usually 2℃-6℃ ). Thus the temperature difference in this study was sufficient to cause adapted regulation of plant functional traits.” (Lines 125-129). This explanation does not satisfy the question. We know perfectly well that the surface temperature corresponding to satellite images is correlated with that of the air. How? You only mention that but not how many degrees, in this way there may be a fault in which the air temperatures are much lower than those of the air, which are important because the trees perceive the air temperatures more than those of the surface making the thermal gradient smaller than that determined with satellite images.
Question: There is also a large amount of literature on altitude changes and the authors did not take them into account, they only dedicated themselves to mentioning literature from this century, as if the 20th century did not exist.In particular.
Response: Regarding the lack of elevation change literature you mentioned, we have added the corresponding references (Lines 471-477). We also list them below:
1. Islam, T.; Hamid, M.; Nawchoo, I.A.; Khuroo, A.A. Leaf functional traits vary among growth forms and vegetation zones in the himalaya. Science of The Total Environment 2024, 906, 167274. Once again 21st century
2. Halbritter, A.H.; Fior, S.; Keller, I.; Billeter, R.; Edwards, P.J.; Holderegger, R.; Karrenberg, S.; Pluess, A.R.; Widmer, A.; Alexander, J.M. Trait differentiation and adaptation of plants along elevation gradients. Journal of Evolutionary Biology 2018, 31, 784-800. Once again 21st century
3. Körner, C.; Neumayer, M.; Menendez-Riedl, S.P.; Smeets-Scheel, A. Functional morphology of mountain plants. Flora 1989, 182, 353-383. This is the only one from 20th century.
Question:It is recommended that the authors include key species of the 15 selected species in the study and discuss the differences and similarities together with the functional groups.
Response: Thank you very much for suggestion on and it is very meaningful to analyze specific response of each species, which could be used to guide the urban planting in the parks. We will also pay more attention on this aspect in the follow-up research. However, in this study, we are more interest in the overall response of most species along the urban-rural gradient. I disagree with this argument because then the objective is not met. What recommendations would be generated? These recommendations would be biased.
Other concerns:
Line 53: Plants are not strategist, plants present mechanisms instead of strategies.
Response: Thank you for pointing out this problem in manuscript. We have changed “strategies” to “trajectories” (Line 57). We apologize for the misleading, but strategy is a frequently-used term in plant ecology that appears in many references (Sharma et al., 2024; Zhang et al., 2023). In order to avoid misleading to the readers, we had tried to reduced the usage of “strategies” except for some proper nouns. (Lines 54, 87, 380, 405). Ok, but is wrong. The word is mechanisms.
Lines 87 and 88: But analyses were just species divided into deciduous and evergreen. Is this analysis sufficient to determine the effect of urbanization on vegetation dynamics?
Response:Thank you very much for your suggestion, this sentence was indeed too much for our results, we had changed them as follows: The insights gleaned from this study will contribute to the prediction of vegetation dynamics of evergreen and deciduous trees during the ongoing urbanization process. (Lines 91-93). Are you sure of that? Prove it!!! Studying functional groups is just a first approximation and can lead to erroneous solutions.
Line 96: What is a heat day?
Response: Thank you so much for your question. According to the criteria of the China Meteorological Administration (daily maximum temperature ≥ 35℃) (https://www.cma.gov.cn), a total of 59 days were classified as extreme hot weather during the study period. The maximum daily temperature in Jinhua City in summer reached 40 ℃, indicating that plants in Jinhua City had experienced extreme high temperature in summer, especially the average monthly temperature in urban areas had reached 37.4℃ (Figure 1f). (Lines 103-109). My question was not answered at all well and I keep asking what is it? Also, the paragraph between the lines 103-109 does not clarify much, 40°C during what time in the day?
Line 103 and 104: Then, this is NOT urban heat island (UHI) but surface urban heat island (SUHI). UHI is by definition air temperature, not surface.
Response:Thank you very much for your suggestion, and I'm sorry for the misleading. Studies have indicated that the land surface temperature (LST) was highly correlated to air temperature (Zhou et al., 2019), we also list it below. Thus, in this study LST was only used to quantify the UHI effects with a high spatial resolution. These data facilitated the generation of detailed spatial distribution maps of UHI effects in Jinhua City, and we have made a supplement in the text. (Lines 114-117).
1. Zhou, D.; Xiao, J.; Bonafoni, S.; Berger, C.; Deilami, K.; Zhou, Y.; Frolking, S.; Yao, R.; Qiao, Z.; Sobrino, J.A. Satellite remote sensing of surface urban heat islands: Progress, challenges, and perspectives. In Remote Sensing, 2019, 11(1), 48-84.
Yes, but you did not consider that the air temperature can be several degrees Celsius lower than the surface temperature. You compared the surface temperature and that is a mistake, and in the discussion section you don't say anything about it.
Line 137: We think it would be better to indicate the sampling sites with numbers instead of abbreviations, in addition to placing them according to the gradient, from the city center to the rural area. With this, readers would have a better understanding of the figures.
Response: We gratefully appreciate for your valuable suggestion. We've replaced the abbreviations of each site with numbers (Figure 1b, c, g and h). (Line 139-143).As showed below:
Table 1:
Line 5 of table 1, Ligustrum lucidum is an evergreen species, is missing
Response: Thank you so much for your careful check. We have added them in the Table. (Line 5 of Table 1).
Line after 6 on table 1, number is missing?
Response: Thank for your comment. The length of the species name is too long, the second half is shown in the next line, we have adjusted the table. (Line after 6 on Table 1).
Line after 7 on table 1: atropurpurea???
Check carefully this table 1
Response: Thank you so much for your careful check. We have carried out a detailed check and adjustment of the form. (Table 1).
Line 141: Specific leaf area (SLA) is missing
Response: We gratefully appreciate for your valuable suggestion. We have already added it in the main text. (Line 169).
Line 142 chlorophyll (Chl)?
Response: Thank you so much for your careful check. The full name of chlorophyll appears in the “Introduction” section, abbreviations are used to avoid repetition. (Line170).
Line 143: stomatal area or stomatal density? The area of a stoma can vary depending on its opening.
Response: Sorry, we changed the stomatal area in the paper to stomatal density. (Line171).
Lines 149 and 150: So in the city-rural gradient the same species was not always studied?
Response: Thanks for your constructive remarks. It should be emphasized that we indeed did not use exactly the same tree species in each sample sites. On the one hand, it is hard to guarantee that all the sites have the same species (you do need the same species in the temperature gradient but not in the whole gradient). On the other hand, when the same species were selected, response of one of few species will probably draw a biased conclusion. Therefore, in order to more truly reflect the response of nature vegetation, dominated species in each regions were selected. This method has been widely adopted in the previous studies to investigate vegetation response under various environmental gradients, such as drought, elevation, precipitation etc. (Line 152-159). The application of elaborate statistic methods have potential to minimize the errors brought by the species differences, as we did in our study. We had illustrated this part in the “data analysis” sections. (Line 222-226).
Line 156: What accuracy?
Response: Thank for your comments, But we did not find the word accuracy in the whole text, we hope you can provide more detailed information for us to make it more clear. Accuracy of the used instrument.
Line 157: What kind of vernier? What about evaporation?
Response: We gratefully appreciate for your valuable suggestion. We have added more detailed information about vernier calipers in the article. We did not find the word “evaporation” in the whole text, we hope you can provide more detailed information for us to make it more clear. (Line 186). When measuring the thickness of a leaf it has to do with evaporation or evapotranspiration
Line 190: Does the data come from a normal population? Did you do any statistical test before applying the ANOVA?
Response: Thank you. To investigate the diverse responses of urban plants to urbanization, we first per-form logarithm or square root transformation on the variables that are not normally distributed to achieve a normal distribution before the analysis. And you check it that using this technique the population was normalized?
Author Response
Many thanks to the reviewer for your questions on our manuscript, which have greatly improved its quality. In this revision, we have made detailed changes according to your comments, and we hope that this revision will meet with your approval.

Round 3
Reviewer 1 Report
Comments and Suggestions for Authors
This new manuscript improved significantly and is more understandable and robust.